# Grounding GUI Anything: Efficient and Semantically-Aware Parsing via Continuous Coordinate Decoding

## Abstract

Recent advances in Multimodal Large Language Models (MLLMs) have substantially improved GUI grounding tasks. However, the following challenges still exist in prior methods: (1) They predict coordinates as discrete tokens in an autoregressive text generation paradigm, which constrains grounding accuracy and leads to sub-optimal inference efficiency; (2) Their predictions are restricted to predefined element sets, and lack the ability to comprehensively parse the entire interface, thereby impeding the versatility and generalizability required for downstream applications. To address these challenges, we introduce **Grounding GUI Anything (GGA)**, an efficient end-to-end framework that enables semantically-aware and fine-grained interface parsing with continuous coordinate decoding. By bridging the MLLM with a dedicated regression-based decoder, the enhanced visual and textual representations are jointly leveraged to regress target coordinates within a continuous spatial domain. This design overcomes the quantization and sequential limitations of traditional discrete token modeling, thus enhancing both localization accuracy and inference speed. Furthermore, to improve robustness and mitigate hallucination, we incorporate a rejection mechanism that enables the model to identify non-existent elements. To facilitate systematic evaluation, we introduce ScreenParse, a comprehensive benchmark designed to assess the structural perception capabilities of GUI grounding models across diverse real-world scenarios. Extensive experiments on ScreenSpot, ScreenSpot-v2, CAGUI-Grounding and ScreenParse benchmarks demonstrate that GGA consistently achieves superior performance compared to existing state-of-the-art methods. All resources will be made publicly available for future research.

## 1 Introduction

GUI-oriented MLLMs are capable of integrating visual and textual data to understand and interact with graphical user interfaces, providing a solid foundation for creating GUI agents. GUI agents have the potential to autonomously operate a wide range of devices, thereby transforming human-computer interaction from an entirely manual process to automated and delegated workflows. Since most of the training data for general MLLMs are natural images, their perception ability on GUI images is insufficient, which limits their effectiveness in GUI-specific contexts. Natural images often contain complex scenes with diverse objects and backgrounds, whereas GUI images are more structured, featuring elements like texts, buttons, and input boxes that have specific functions and layouts. For GUI scenario perception and advanced tasks, ideal GUI MLLMs not only need to understand the ever-changing and high information-density interfaces on various devices, but also exactly perform basic operations, such as understanding the semantics of interface elements and outputting precise coordinates.

To improve the GUI perception, recent works (Cheng et al., 2024; Xu et al., 2024; Gou et al., 2024) attempt to employ pre-trained MLLMs as backbone. Specifically, they almost all adopt textual autoregressive modeling to generate coordinates as a sequential stream of discrete tokens, as presented in Figure 1 (left). Although these methods have made some progress in GUI perception, they also have some limitations. **(1)** Generating coordinates as a sequence of discrete tokens fails to holistically perceive the spatial configuration of GUI elements and tends to yield sub-optimal

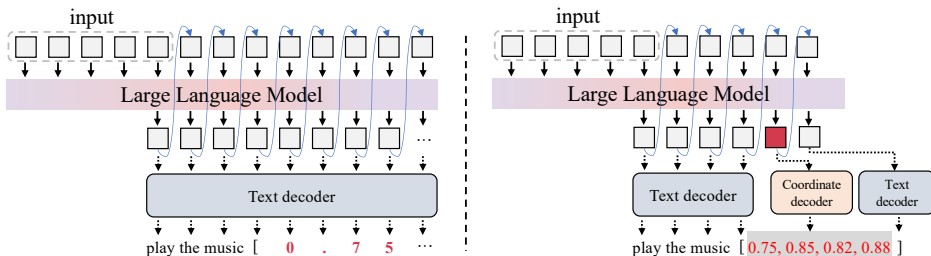

Figure 1: **Comparison of the coordinate generation between prior methods (left) and ours (right)**. We utilize features instead of multiple discrete tokens to obtain continuous coordinate values, thereby improving the precision of grounding and speeding up the inference. In fact, a special [VG] token (the red token) represents coordinates of a bounding box in our method.

localization accuracy. The reason is that this autoregressive mechanism is originally designed for natural language generation and the discrete tokens can hardly represent the geometric information of localization. Besides, token-by-token prediction manner of high-precision coordinates could be a bottleneck when rapid response time is required. **(2)** Previous models may return incorrect locations or generate irrelevant responses when a non-existent element is required to be located, leading to failures in downstream tasks, which can significantly impact user experience and system reliability, as illustrated in Figure 2 (left). **(3)** They only focus on locating and interacting with specified elements and ignore providing a detailed parsing perception[1] of broader context and relationships between elements on the entire user interface, which can refer to Figure 2 (right). To a certain extent, these three points limit the development of a more powerful and robust GUI MLLM.

To address the aforementioned limitations, we propose a novel end-to-end model, called **Grounding GUI Anything (GGA)**, which can achieve robust grounding of specified elements and detailed parsing of the entire user interface within a model simultaneously. Specifically, we design a route-then-predict framework to efficiently process both vision and language information, consisting of an MLLM, a token router, a vision adapter, a coordinate decoder, and an additional element matcher for the training phase only. The output tokens from the MLLM are classified by the token router into text tokens and visual grounding tokens. Text tokens are decoded into element semantics, while visual grounding tokens, combined with visual features from the vision adapter, are processed by the coordinate decoder for localization. Instead of treating the coordinates as multiple discrete tokens in previous GUI models, the lightweight coordinate decoder adds a special [VG] token. And we combine the special token and the continuous characteristic of image space to provide higher-precision grounding ability and more efficient inference, as shown in Figure 1(right). Then, the element matcher is utilized to ensure that element semantics from the instruction and candidate coordinates are correctly matched for multi-target grounding and parsing of user interfaces. Besides, we also introduce [REJ] token to represent elements that do not exist in the user interface. When [REJ] tokens appear, their coordinate decoding process is skipped straightforwardly, avoiding unnecessary computations and hallucination. To provide a detailed evaluation of our method, we construct a large-scale GUI parsing dataset, ScreenParse, which consists of 500K training samples. Leveraging self-constructed parsing datasets, we train our model for parsing to extract a comprehensive representation of all elements, including their semantics and corresponding locations, empowering our model with parsing capability on the entire user interface. Specifically, we conduct extensive evaluation on ScreenSpot, ScreenSpot-v2, CAGUI-Grounding, and ScreenParse benchmarks, and our proposed GGA consistently achieves superior performance compared to state-of-the-art methods.

To summarize, our contributions are listed as follows: **(1)** We propose a novel end-to-end model Grounding GUI Anything (GGA), which can realize robust grounding of specified elements and detailed parsing of the entire user interface within a model simultaneously by adding specialized [REJ] and [VG] tokens. **(2)** We compress multiple discrete coordinate tokens into a single specialized [VG] token and use coordinate decoder for combining vision feature and special token to substantially shorten inference time, which can also improve the accuracy for coordinate-related tasks, thereby benefiting a wider range of downstream applications. **(3)** To provide a comprehensive

---

[1]The parsing task means that the model grounds all elements in the GUI interface, including icons and texts.

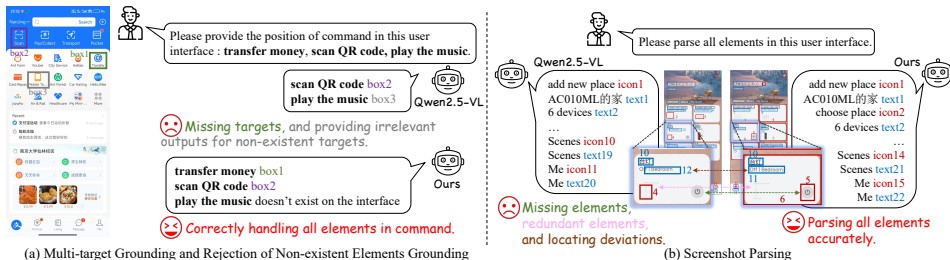

(a) Multi-target Grounding and Rejection of Non-existent Elements Grounding

(b) Screenshot Parsing

Figure 2: **Qualitative illustration.** Our method delivers robust predictions with high grounding precision, whereas Qwen2.5-VL exhibits missing values, redundant elements and locating deviations.

evaluation, we construct a large-scale GUI parsing dataset, ScreenParse, consisting of 500K samples. We also conduct extensive experiments on ScreenSpot, ScreenSpot-v2, CAGUI-Grounding and ScreenParse benchmarks to verify the superior performance of our proposed GGA.

## 2 RELATED WORKS

### 2.1 GENERAL MLLMS

Recent years have witnessed rapid progress in General MLLMs. GPT-4V (Yang et al., 2023) and Gemini (Team et al., 2023) boosted performance via massive web-scale data and reinforcement learning from human feedback, achieving strong zero-shot visual understanding and reasoning. Later, open-source models such as LLaVA (Liu et al., 2023), Qwen-VL (Bai et al., 2023) and InternVL (Chen et al., 2024c) leveraged instruction-tuning on curated datasets to align LLMs with visual inputs efficiently. They fused vision encoders and language decoders end-to-end, supporting diverse modalities including audio and video. Despite these advances, general MLLMs are mainly trained on natural images rather than GUI images, exhibiting limited perceptual capability in GUI-specific scenarios.

### 2.2 MLLMS FOR GUI GROUNDING

To achieve GUI grounding tasks, several representative works (Cheng et al., 2024; Gou et al., 2024; Wu et al., 2024; Yang et al., 2024; You et al., 2024; Xu et al., 2024; Qin et al., 2025) have explored to fine-tune MLLMs with large-scale text-position pairs extracted from user interfaces. Seeclick (Cheng et al., 2024) achieved automatic GUI perception solely based on user interfaces for the first time, and proposed a multi-platform benchmark to evaluate GUI grounding. Ferret-UI series (You et al., 2024; Li et al., 2024) used a dynamic resolution strategy to magnify the interface details to enhance visual perception. Aguvis (Xu et al., 2024) and UI-TARS (Qin et al., 2025) collected a large amount of annotated data and utilized reasoning paradigms to achieve stronger GUI perception capabilities. To fully utilize the semantic and location information of elements in the user interface, OmniParser (Wan et al., 2024) leveraged powerful expert models to extract icons and texts, after which GPT-4V (Yang et al., 2023) was applied to generate the functionality or semantics of each element. Although they have made some progress, the following issues still exist: (**1**) Their way of modeling coordinates is discrete and sequential prediction, sacrificing both grounding precision and inference speed. (**2**) When an element does not exist on the interface, they generate false responses (grounding hallucination) rather than correct rejections. (**3**) They cannot achieve parsing on the entire interface or require the use of additional tools to complete parsing in a non-end-to-end method. We propose an end-to-end and continuous modeling of coordinates framework to enable parsing all elements from a user interface and deal with non-existent elements correctly, providing a high-precision, efficient and effective perception of GUI interfaces.

## 3 METHOD

To develop the ability of MLLM-based GUI perception, we introduce **Grounding GUI Anything** (**GGA**), an end-to-end framework capable of robustly localizing specified elements and compre-

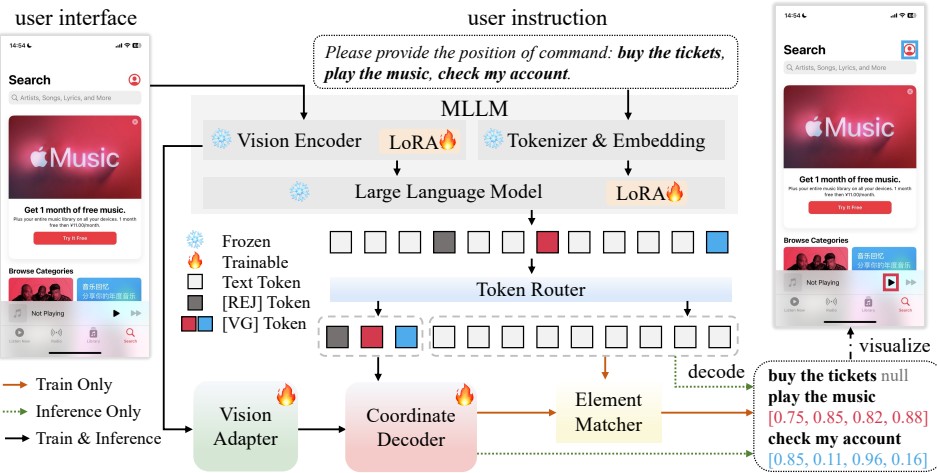

Figure 3: **Overall architecture of Grounding GUI Anything (GGA)**. A token router first classifies output tokens of the MLLM into text and special tokens ([VG] and [REJ]). Text tokens are handled by the MLLM. Features of [VG] tokens and vision-adapter features are jointly decoded for exact coordinates, while [REJ] tokens are discarded. An element matcher aligns predicted elements with ground truth during training to avoid the variance in the generated element order.

hensively parsing the entire user interface simultaneously. The overall architecture of **Grounding GUI Anything (GGA)** is illustrated in Figure 3. Specifically, the model adopts a route-then-predict architecture that processes vision and language streams through an MLLM, a token router, a vision adapter, a coordinate decoder, and an element matcher used only during training. Tokens produced by the MLLM are separated into textual tokens and visual grounding tokens by the token router. Textual tokens are decoded into element semantics, whereas visual grounding tokens, combined with image features from the vision adapter, are input to a lightweight coordinate decoder. Then the decoder regresses continuous coordinates via special [VG] tokens, yielding higher localization accuracy and improved inference efficiency compared to discrete token generation. During training, the element matcher ensures correct alignment between predicted semantics and candidate coordinates, enabling multi-target grounding and parsing. In addition, [REJ] tokens are introduced to represent non-existent elements, allowing their coordinates to be skipped during decoding, which reduces unnecessary computation and suppress hallucination.

## 3.1 MODEL ARCHITECTURE

**MLLM.** The MLLM serves as the backbone for jointly processing user instructions and GUI interfaces, providing unified semantic representations for subsequent grounding and parsing tasks. The user instruction $T_{instruction}$ and the interface $I$ are first processed by an fine-tuned MLLM $\mathcal{F}_{MLLM}$ to obtain semantic features $f_{token}$ as

$$f_{token} = \mathcal{F}_{MLLM}(I, T_{instruction}). \tag{1}$$

**Token Router.** To decouple spatial grounding from token-by-token generation, we introduce a token router that partitions the MLLM-generated tokens into semantic tokens and localization tokens. The routing decision is based on token-level logits to identify two categories of localization tokens: [VG] tokens indicating bounding boxes of target elements, and [REJ] tokens representing non-existent targets. [REJ] tokens are discarded to prevent unnecessary coordinate computation, while [VG] tokens proceed to the coordinate decoding stage. Semantic tokens are handled via the MLLM's standard autoregressive text generation to produce human-readable descriptions of interface elements.

**Vision Adapter.** Instead of relying on a large-scale external vision encoder to augment the model's perception, we leverage the MLLM's internal vision encoder $\mathcal{F}_{ViE}$ and introduce a lightweight, task-specific vision adapter $\mathcal{F}_{adapter}$. This adapter fine-tunes the original visual representations for GUI-oriented localization, producing enhanced visual features $f_{vision}$ as:

$$f_{vision} = \mathcal{F}_{adapter}(\mathcal{F}_{ViE}(I)). \tag{2}$$

By aligning the adapted visual features with textual reasoning in the MLLM, this design provides efficient visual-semantic integration while retaining consistency with the backbone's latent space.

**Coordinate Decoder.** To overcome the grounding quantization and inference efficiency limitations of token-by-token coordinate generation in existing GUI MLLMs, we introduce a lightweight coordinate decoder for direct regression of continuous values. For each [VG] token, we extract its final-layer embedding from the MLLM as the text feature, and pair it with the GUI-specific visual features from the vision adapter. The coordinate decoder $\mathcal{F}_{decoder}$ then outputs the bounding box coordinates $\mathcal{O}_{BBox}$ via:

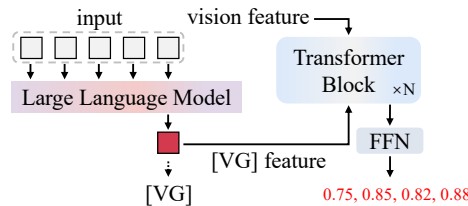

Figure 4: **Illustration of continuous coordinate decoding in Grounding GUI Anything.**

$$\mathcal{O}_{BBox} = \mathcal{F}_{decoder}(f_{token_{[VG]}}, f_{vision}). \tag{3}$$

This approach bridges the discrete-to-continuous gap in coordinate-related tasks: **(1)** Higher precision: avoiding quantization errors inherent in vocabulary-based coordinate prediction. **(2)** Computational efficiency: compressing multiple coordinate tokens into a single special token, substantially reducing inference latency for both grounding and parsing tasks. More details of the Coordinate Decoder are presented in Appendix.

**Element Matcher.** We adopt an element matching strategy that integrates semantic similarity and bounding-box IoU into a unified score, enabling alignment of predicted and ground-truth elements based on both meaning and spatial consistency. This ensures each ground-truth element is paired with its nearest predicted counterpart, yielding two benefits: **(1)** The loss is computed only between semantically and spatially aligned pairs, avoiding penalization for output ordering. **(2)** The model gains robustness to diverse interface layouts and element arrangements, improving generalization to real-world GUI scenarios. Further details of the matching algorithm are provided in Section 3.2.

## 3.2 TRAINING OBJECTIVES

To train the framework on complex tasks with multiple types of instructions rather than just a simple grounding task, we elaborately design a set of objectives to ensure stable convergence during training. The output could be multiple elements with their corresponding bounding boxes and there might exist inconsistency between the output order and ground-truth element order of elements. In contrast to closed-set matching in DETR (Carion et al., 2020), which aligns predicted boxes with ground-truth via classification scores and spatial overlap, we adopt an open-form matching strategy tailored for GUI grounding in MLLMs. Specifically, predicted elements are paired with ground-truth annotations by jointly considering semantic similarity and bounding-box IoU within a unified assignment objective, followed by optimization of the corresponding element-wise losses.

Let $y = \{(t_i, b_i)\}$ denote the ground-truth set of elements and $\hat{y} = \left\{(\hat{t}_i, \hat{b}_i)\right\}$ the set of predictions. $t_i$ and $\hat{t}_i$ are the semantic feature of the text for the $i^{th}$ element, while $b_i$ and $\hat{b}_i$ represents its bounding box. To find a match for ground truth element $y_i$, we search the set of predictions using a matching cost:

$$\sigma = \begin{cases} \underset{j}{\arg\min} \; \mathcal{C}_{match}(y_i, \hat{y}_j) & \text{if } \mathcal{C}_{match}(y_i, \hat{y}_j) > \mu, \\ \emptyset & \text{otherwise}, \end{cases} \tag{4}$$
$$\text{where } \mathcal{C}_{match} = \lambda_{IoU}\mathcal{L}_{IoU}(b_i, \hat{b}_j) + \lambda_{sem}\mathcal{L}_{sem}(t_i, \hat{t}_j),$$

where $\mu$ is the threshold to evaluate whether two elements match. And $\mu$ is experimentally set to 0.55. After finding a match for each ground truth element, we use cross-entropy loss for the text, Smooth-L1 loss and IoU loss (Rezatofighi et al., 2019) for the bounding box:

$$\mathcal{L}_i = \lambda_{CE}\mathcal{L}_{CE}(t_i, \hat{t}_\sigma) + \lambda_{L1}\mathcal{L}_{L1}(b_i, \hat{b}_\sigma) + \lambda_{IoU}\mathcal{L}_{IoU}(b_i, \hat{b}_\sigma). \tag{5}$$

Since each element contains semantic and coordinate optimization, for a ground truth set with $N_{gt}$ elements the total loss is defined as:

$$\mathcal{L} = \sum_{i}^{N_{gt}} \mathcal{L}_i \tag{6}$$

These training objectives help with more stable learning from complex tasks with an indefinite number of elements. The model can better handle the challenges posed by varying numbers of elements in different user interfaces, leading to improved performance and robustness.

# 4 BENCHMARK

## 4.1 DATA COLLECTION AND ANNOTATION

We construct a parsing benchmark spanning diverse domains and application scenarios to ensure variability in interface design, functionality, and complexity. Building upon the open-source ScreenSpot benchmark (Cheng et al., 2024), we re-annotate all interface instances with element types, semantic labels, and bounding boxes. To further broaden the coverage of user interface styles and languages, we additionally annotate a collection of interfaces from widely used Chinese applications using the same annotation protocol. The detailed statistics of our proposed ScreenSpot can be found in Appendix.

We construct the benchmark using a semi-automatic pipeline that combines expert models with manual verification for quality control. Icon elements are identified using the open-domain detector Grounding DINO (Liu et al., 2024b), and textual elements are extracted via PaddleOCR (Authors, 2020). Redundant bounding boxes are removed using non-maximum suppression (NMS), and missing semantics are supplemented by a pre-trained MLLM. All annotations are manually reviewed to correct potential errors in both bounding boxes and semantic labels. The dataset comprises 1000 English and Chinese interfaces in equal proportion (500 each), with an average of 36 annotated elements per interface, and the annotation pipeline along with examples is illustrated in Figure 5.

## 4.2 EVALUATION METRICS

**Bounding Box Performance.** Two complementary metrics are employed to evaluate bounding-box performance in GUI parsing: element recall, defined as the proportion of ground-truth elements correctly localized, and element precision, defined as the proportion of predicted elements that correspond to ground-truth instances. High recall reflects the model's capacity to detect the majority of existing elements, whereas high precision indicates that most localized elements are correct, thereby reducing false positives.

**Semantics Performance.** Semantic Similarity (SemSim) is introduced to evaluate the alignment between predicted and ground-truth element semantics for matched bounding boxes, encompassing both textual and icon

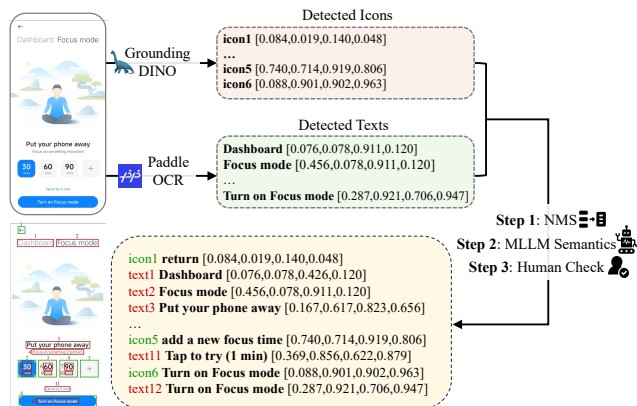

Figure 5: **Semi-automatic pipeline and examples of data annotation in ScreenParse.** Element types, semantics, bounding boxes for all elements are annotated.

descriptions. This metric reflects the model's capability to generate semantically faithful descriptions of localized elements, which is essential for tasks requiring deep GUI comprehension, such as intelligent user interaction and context-aware assistance.

Table 1: **Quantitative comparison on ScreenSpot and ScreenSpot-v2**. The **best** and second best performance are marked, where "-" indicates that the information cannot be obtained due to missing values or API usage and "∗" means that additional private datasets of undisclosed size were used.

| Model | # Data | Mobile | | Desktop | | Web | | Avg. | Time(s) |
|-------|--------|--------|------|---------|------|------|------|------|---------|
| | | Text | Icon | Text | Icon | Text | Icon | | |
| *ScreenSpot* | | | | | | | | | |
| GPT-4o | - | 22.6 | 24.5 | 20.2 | 11.8 | 9.2 | 8.8 | 18.3 | - |
| Claude Computer Use | - | - | - | - | - | - | - | 83.0 | - |
| Gemini 2.0 (Project Mariner) | - | - | - | - | - | - | - | 84.0 | - |
| Qwen2.5-VL-7B | - | **97.1** | 81.2 | 86.6 | 70.0 | 87.4 | 78.6 | 84.9 | 0.763 |
| InternVL2.5-8B | - | 82.8 | 58.5 | 47.4 | 28.6 | 47.4 | 26.7 | 48.6 | 0.812 |
| CogAgent-18B | 400K | 67.0 | 24.0 | 74.2 | 20.0 | 70.4 | 28.6 | 47.4 | 1.112 |
| SeeClick-9.6B | 1M | 78.0 | 52.0 | 72.2 | 30.0 | 55.7 | 32.5 | 53.4 | 0.872 |
| OmniParser | - | 93.9 | 57.0 | 91.3 | 63.6 | 81.3 | 51.0 | 73.0 | - |
| UGround-7B | 1.3M | 82.8 | 60.3 | 82.5 | 63.6 | 80.4 | 70.4 | 73.3 | 0.821 |
| OS-Atlas-7B | 2.3M | 93.0 | 72.9 | 91.8 | 62.9 | 90.9 | 74.3 | 82.5 | 0.782 |
| Aguvis-7B | 1M | 95.6 | 77.7 | 93.8 | 67.1 | 88.3 | 75.2 | 84.4 | 0.791 |
| GUI-Actor-7B | 9.6M | 94.9 | 82.1 | 91.8 | 80.0 | 91.3 | **85.4** | 88.3 | 0.797 |
| UI-TARS-7B | 18.4M∗ | 94.5 | 85.2 | **95.9** | **85.7** | 90.0 | 83.5 | 89.5 | 0.741 |
| **GGA-8B** | 515K | 96.9 | **86.8** | **95.9** | 85.3 | 91.4 | 84.6 | **90.2** | **0.173** |
| *ScreenSpot-v2* | | | | | | | | | |
| OpenAI Operator | - | 47.3 | 41.5 | 90.2 | 80.3 | 92.8 | 84.3 | 70.5 | - |
| GPT-4o + OmniParser-v2 | - | 95.5 | 74.6 | 92.3 | 60.9 | 88.0 | 59.6 | 80.7 | - |
| Qwen2.5-VL-7B | - | **99.0** | 84.4 | 87.6 | 65.7 | 90.2 | 79.8 | 86.5 | 0.756 |
| InternVL2.5-8B | - | 84.5 | 58.3 | 48.5 | 30.0 | 43.6 | 27.1 | 48.7 | 0.793 |
| CogAgent-18B | 400K | 69.3 | 27.0 | 75.8 | 20.7 | 74.4 | 31.5 | 52.8 | 1.104 |
| SeeClick-9.6B | 1M | 78.4 | 50.7 | 70.1 | 29.3 | 55.2 | 32.5 | 55.1 | 0.864 |
| UGround-7B | 1.3M | 84.5 | 61.6 | 85.1 | 61.4 | 84.6 | 71.9 | 76.3 | 0.805 |
| OS-Atlas-7B | 2.3M | 95.2 | 75.8 | 90.7 | 63.6 | 90.6 | 77.3 | 84.1 | 0.774 |
| Aguvis-7B | 1M | 95.5 | 81.5 | 93.3 | 77.9 | 91.0 | 77.8 | 87.3 | 0.779 |
| UI-TARS-7B | 18.4M∗ | 96.9 | 89.1 | 95.4 | 85.0 | 93.6 | 85.2 | 91.6 | 0.733 |
| Jedi-7B | 1.3M | 96.9 | 87.2 | 95.9 | **87.9** | 94.4 | 84.2 | 91.7 | 0.763 |
| **GGA-8B** | 515K | 98.5 | **90.6** | **96.1** | 85.5 | **94.8** | **86.6** | **92.0** | **0.168** |

## 5 EXPERIMENTS

### 5.1 EXPERIMENTAL SETUPS

**Datasets.** Our training data is composed of open-source English datasets and self-annotated Chinese datasets, including 515k training samples in the fields of mobile, desktop and web (See more details in Section A.1). Furthermore, to optimize training efficiency and enhance model robustness, we implemented a multi-target instruction paradigm, wherein each training instance randomly encompasses multiple objectives, departing from the conventional single-target approach.

**Baselines.** We compare the proposed GGA with various baselines, including closed-source commercial models GPT-4o (Hurst et al., 2024), Claude Computer Use (Anthropic, 2024), Gemini 2.0 (Project Mariner) (GoogleDeepmind, 2024), as well as open-source basic models Qwen2.5-VL series (Bai et al., 2025), InternVL2.5 series (Chen et al., 2024b) and academic GUI models SeeClick (Cheng et al., 2024), OmniParser series (Wan et al., 2024; Yu et al., 2025), CogAgent (Hong et al., 2024), AgentCPM-GUI (Zhang et al., 2025), Aguvis (Xu et al., 2024), OS-Atlas (Wu et al., 2024), UGround (Gou et al., 2024), GUI-Actor (Wu et al., 2025), Jedi (Xie et al., 2025) and UI-TARS (Qin et al., 2025).

**Implementation details.** In our work, we use InternVL2.5-8B as the backbone $\mathcal{F}_{MLLM}$, and the vision adapter $\mathcal{F}_{adapter}$ is an MLP module with channels $[4096, 2048, 1024, 256]$. The coordinate decoder is a transformer-based architecture with cross-attention mechanism where the dimension of hidden states is 256, and it finally regresses to the coordinates through a decoding head. We adopt 8 NVIDIA 80G A100 to train the whole framework, and the vision encoder is unfrozen during training. LoRA (Hu et al., 2022) is applied to fine-tune both vision and language parts in MLLM, with lora-rank $r$ and lora-alpha $\alpha$ set to 16 and 32. We also adopt CosineLRScheduler as the learning rate scheduler, where the learning rate and warmup ratio are set to $3e^{-5}$ and 0.03 respectively. In the training process, $\lambda_{IoU}$ and $\lambda_{sem}$ in element matcher are both set to 1.0 in Eq. 4, while $\lambda_{CE}$, $\lambda_{L1}$ and $\lambda_{IoU}$ in Eq. 5 are set to 2.0, 4.0 and 1.0 respectively. More details can be found in Appendix.

Table 2: **Quantitative comparison on CAGUI-Grounding.** The **best** and second best performance are marked. The results demonstrate that our method exhibits strong multilingual GUI grounding performance.

| Model | Func2Box | Text2Box | Avg. | Time(s) |
|---|---|---|---|---|
| GPT-4o | 22.1 | 19.9 | 21.0 | - |
| GPT-4o (grounding) | 44.3 | 44.0 | 44.2 | - |
| Qwen2.5-VL-7B | 59.8 | 59.3 | 59.6 | 2.512 |
| InternVL2.5-8B | 17.2 | 24.2 | 20.7 | 2.684 |
| OS-Atlas-7B | 53.6 | 60.7 | 57.2 | 2.542 |
| UI-TARS-7B | 56.8 | 66.7 | 61.8 | 2.373 |
| Aguvis-7B | 60.8 | 76.5 | 68.7 | 2.578 |
| AgentCPM-GUI-8B | 79.1 | 76.5 | 77.8 | 2.691 |
| **GGA-8B** | **83.2** | **82.9** | **83.1** | **0.632** |

Table 3: **Quantitative comparison on ScreenParse.** The **best** and second best performance are marked, where "-" indicates the time consumption cannot be obtained because of API usage.

| Model | English | | | Chinese | | | Time(s) |
|---|---|---|---|---|---|---|---|
| | Recall | Precision | SemSim | Recall | Precision | SemSim | |
| GPT-4o | 5.7 | 11.7 | 0.586 | 4.9 | 7.8 | 0.406 | - |
| Claude Computer Use | 17.1 | 35.3 | 0.758 | 19.3 | 31.0 | 0.511 | - |
| Qwen2.5-VL-7B | 18.8 | 36.6 | 0.596 | 25.0 | 52.6 | 0.854 | 0.534 |
| Qwen2.5-VL-72B | 22.9 | 43.7 | 0.685 | 42.4 | 70.5 | 0.867 | 2.748 |
| InternVL2.5-8B | 11.7 | 30.4 | 0.615 | 28.6 | 47.9 | 0.812 | 0.558 |
| InternVL2.5-78B | 20.4 | 38.6 | 0.628 | 40.8 | 66.4 | 0.836 | 2.822 |
| Jedi-7B | 34.7 | 46.9 | 0.721 | 7.8 | 12.6 | 0.578 | 0.568 |
| **GGA-8B** | **77.2** | **77.9** | **0.918** | **87.1** | **89.5** | **0.946** | **0.154** |

## 5.2 MAIN RESULTS

**Quantitative Comparison of Grounding.** We evaluate GUI grounding on three benchmarks: (1) ScreenSpot: a single-step grounding benchmark covering multiple platforms; (2) ScreenSpot-v2 (Wu et al., 2024): a re-annotation version based on ScreenSpot that corrects some coordinate and instruction errors; (3) CAGUI-Grounding (Zhang et al., 2025): a Chinese mobile text and function grounding evaluation benchmark. GGA employs a transformer-based coordinate decoder to fuse visual and textual features for direct continuous coordinate regression, outperforming autoregressive baselines in both accuracy and efficiency. As shown in Table 1 and Table 2, GGA achieves state-of-the-art results while using a substantially smaller dataset, delivers the fastest inference speed, and exhibits strong adaptability to multilingual interfaces and varied instruction types.

**Quantitative Comparison of Parsing.** To evaluate the model's global perception of both the semantics and spatial coordinates of user interface elements, we evaluate on the proposed ScreenParse benchmark, which contains English and Chinese interfaces. Average inference time per predicted element is measured using the *transformer* framework on an NVIDIA A100 (80G). As shown in Table 3, GGA delivers superior parsing performance across diverse interfaces. In contrast, commercial and other large-scale models detect only a limited set of elements, resulting in lower recall and precision and reduced semantic similarity, indicating deviations in their understanding of element semantics. By decoupling the semantic and spatial components of the parsing task and incorporating an element matcher to ensure correct alignment during training, our approach achieves substantial gains in accuracy and recall over baselines, while markedly reducing per-element inference time.

**Analysis of Hallucination Suppression.** Previous GUI grounding methods typically localize a single or few queried elements by directly generating bounding boxes, without a global verification mechanism, making them prone to hallucination where plausible but non-existent elements are predicted. In contrast, Grounding GUI Anything (GGA) achieves parsing the semantics and coordinates of all present elements for the first time, enabling explicit existence checks before localization. To evaluate hallucination suppression, we construct a hallucination-sensitive split from ScreenParse by sampling queries for non-existent elements and measure performance using Rejection Accuracy (RA). The split contains 1,000 query–GUI pairs evenly divided between English and Chinese interfaces. As shown in Table 4, GGA achieves the highest RA, demonstrating that comprehensive parsing significantly mitigates grounding hallucinations in MLLMs.

Table 4: **Quantitative comparison of hallucination suppression.** Our method attains the highest Rejection Accuracy, suggesting that the incorporation of global semantically-aware parsing is instrumental in suppressing hallucination for GUI grounding.

| Model | Rejection Accuracy (RA) |
|---|---|
| GPT-4o | 56.8 |
| Claude Computer Use | 59.4 |
| Qwen2.5-VL-7B | 63.4 |
| InternVL2.5-8B | 60.1 |
| UGround-7B | 8.7 |
| OS-Atlas-7B | 9.8 |
| Aguvis-7B | 10.2 |
| GUI-Actor-7B | 21.5 |
| UI-TARS-7B | 16.9 |
| Jedi-7B | 30.7 |
| **GGA-8B** | **80.2** |

Table 5: **Ablation study.** For conciseness, we primarily focus on results on ScreenSpot-v2 and ScreenParse. Values in parentheses indicate the percentage change relative to the original model.

| Model | ScreenSpot-v2 | | ScreenParse | | |
|---|---|---|---|---|---|
| | Text Avg. | Icon Avg. | Recall Avg. | Precision Avg. | SemanticSim Avg. |
| **full model** | **96.5** | **87.6** | **82.2** | **83.7** | **0.932** |
| w/o coordiante decoder | 91.9 (**-4.8%**) | 82.2 (**-6.2%**) | 77.6 (**-5.6%**) | 78.2 (**-6.6%**) | 0.930 (**-0.2%**) |
| w/o vision adapter | 22.1 (**-77.1%**) | 17.8 (**-79.7%**) | 5.4 (**-93.4%**) | 4.6 (**-94.5%**) | 0.915 (**-1.8%**) |
| w/o element matcher | 95.7 (**-0.8%**) | 87.1 (**-0.6%**) | 72.5 (**-11.8%**) | 73.4 (**-12.3%**) | 0.814 (**-12.7%**) |
| w/o parsing dataset | 96.0 (**-0.5%**) | 87.0 (**-0.7%**) | 20.2 (**-75.4%**) | 39.2 (**-53.2%**) | 0.751 (**-19.4%**) |

## 5.3 ABLATION STUDY

**Impact of Coordinate Decoder.** The second row of Table 5 shows that our decoupled framework surpasses the autoregressive paradigm across benchmarks, corroborating that modeling coordinates through continuous spatial features regression enables more effective coordinate-related process in MLLMs.

**Impact of Vision Adapater.** The third row of Table 5 reveals that removing the visual adapter and relying solely on text features with self-attention for coordinate regression leads to a substantial accuracy drop. This highlights the necessity of rich visual–feature interactions for effective localization of user interface elements.

**Impact of Element Matcher.** The fourth row of Table 5 shows that removing the element matcher reduces parsing accuracy, largely due to the inherent randomness of MLLM generation and target mismatches. This underscores the element matcher's importance in aligning model outputs with the ground truth, thereby alleviating the need for strictly consistent output ordering.

**Impact of Parsing Dataset.** Incorporating an annotated parsing dataset markedly strengthens the model's global structural perception. As shown in the fifth row of Table 5, this addition yields substantial gains in parsing GUI interfaces, while also bringing slight improvements to overall GUI grounding performance.

## 6 CONCLUSION

In this paper, we propose **Grounding GUI Anything (GGA)**, a route-then-predict framework that decouples the semantic understanding and spatial localization of GUI elements. The framework employs a specialized coordinate decoder that directly regresses locations in a continuous feature space rather than serializing them into discrete coordinate tokens via autoregressive text decoding. This design yields state-of-the-art performance across GUI grounding and parsing benchmarks and achieves the highest inference efficiency. Meanwhile, a rejection mechanism is introduced to enhance robustness against grounding hallucination. Furthermore, we present ScreenParse, a comprehensive parsing benchmark that systematically evaluates models' ability to predict the semantics and locations of all GUI elements, providing a foundation for future research on global structural perception in MLLMs.

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

# A    APPENDIX

## A.1    DATA COMPOSITION DETAILS

Our training data consists of two parts, which is shown in Table 6. One part is an aggregated open-source English dataset. Due to the huge amount of such data, we ultilized a deduplication filtering method based on image and instruction similarity. Ultimately, we obtained 400K samples covering various platforms in the following dataset: SeeClick (Cheng et al., 2024), AMEX (Chai et al., 2024), AndroidWorld (Rawles et al., 2024), UIBERT (Bai et al., 2021), FineWeb (Penedo et al., 2024), MultiUI (Liu et al., 2024a), OS-Atlas (Wu et al., 2024), GUIEnv (Chen et al., 2024a) and AutoGUI (Li et al., 2025), as well as a parsing dataset annotated on 5K interfaces. The other part is to annotate 100K grounding and 10K parsing training samples on various Chinese interfaces we collected.

Table 6: Overview of the utilized training datasets in Grounding GUI Anything, where "∗" indicates the open-source dataset.

|  | Grounding | | Parsing | | Total |
|---|---|---|---|---|---|
|  | **English** | **Chinese** | **English** | **Chinese** |  |
| **# Samples** | 400K* | 100K | 5K | 10K | 515K |

## A.2    ARCHITECTURE OF COORDINATE DECODER

The coordinate decoder in our approach is implemented as a transformer-based module, which ingests multimodal features and performs internal interactions to regress continuous coordinate values. The architecture detail of coordinate decoder is shown in Figure 6. In contrast to text autoregressive generation form, our method transforms discrete tokens into continuous coordinate representations and fully exploits multimodal information during the localization stage, thereby enhancing grounding accuracy.

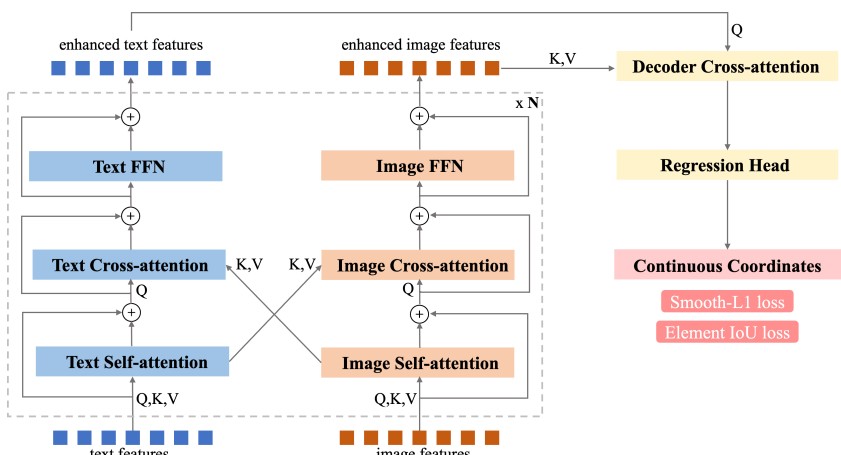

Figure 6: **Architecture of the proposed coordinate decoder.** Hidden text features from the LLM part and enhanced image features from the vision adapter are fused via multi-stage self- and cross-attention to produce updated multimodal representations. A decoder cross-attention module then maps these representations into continuous coordinate space through a box head, supervised with Smooth-L1 and IoU loss. By regressing coordinates directly in a continuous feature space—rather than serializing them into discrete coordinate tokens via autoregressive text generation—this design improves localization accuracy and significantly reduces inference latency.

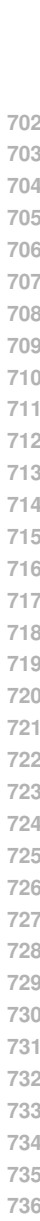

Figure 7: **Convergence curves of evaluation metrics with respect to the number of training steps.** Compared with training strategy using text autoregressive generation on the same basic model, our method has faster convergence in the second stage and higher Recall/Precision performance.

### A.3 TWO-STAGE TRAINING PROCESS

In our proposed method, we employ a two-stage progressive training strategy:

- During the first stage, training is restricted exclusively to grounding task, enabling the model to develop robust GUI localization capabilities, while simultaneously optimizing a randomly initialized grounding decoder.
- During the second stage, training is conducted using both a subset of the grounding annotations and the complete parsing dataset. This joint optimization further enhances the model's GUI perception capabilities.

Our empirical analysis reveals that, since the grounding decoder has been sufficiently optimized during the first training phase, the model primarily concentrates on acquiring the parsing-specific generation patterns during the second stage, eliminating the necessity for substantial additional optimization of the grounding decoder. This approach leads to faster convergence compared to the text autoregressive paradigm, and the convergence comparison of evaluation metrics with the number of training steps is shown in Figure 7.

### A.4 INSTRUCT EXISTING MODELS FOR PARSING

For existing open-source models, we employed various prompts to stimulate their interface parsing capabilities, and all evaluation results are reported based on the corresponding best-performing prompts. However, as a result of extensive fine-tuning on domain-specific datasets, some open-source academic models have exhibited diminished responsiveness to parsing instructions and are generally unable to generate parsing outputs. The prompt template tried to utilize to assess the models' parsing capabilities is presented as follows.

---

**Parsing Prompt Example 1:** Parse the semantics of all elements on the interface and their corresponding box coordinates in the format of [[x1,y1,x2,y2]], including all icons and texts.

**Parsing Prompt Example 2:** Parse the short description of all elements (including icons and texts) on the interface and their corresponding coordinates. The format of the coordinates is [[x1,y1,x2,y2]]. The following is an example output:
icon: play music [[x1,y1,x2,y2]]
text: search for my favorites [[x1,y1,x2,y2]]
text: Today's Hot Songs [[x1,y1,x2,y2]]
icon: play the next song [[x1,y1,x2,y2]]

---

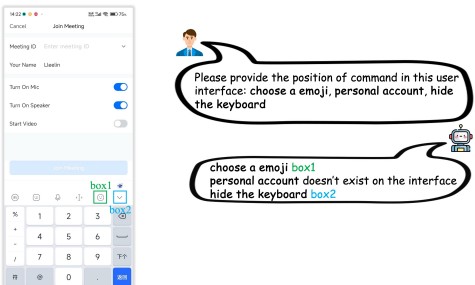
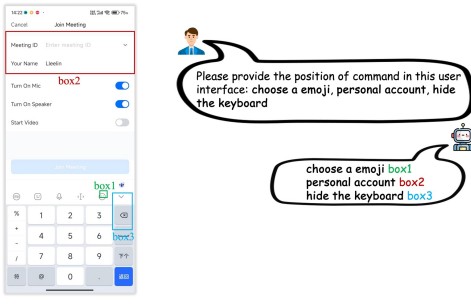

(a) For English interface, Grounding GUI Anything achieves accurate multi-target grounding accuracy and rejection.

(b) For English interface, Qwen2.5-VL-7B showed locating deviations and false positives of non-existent elements.

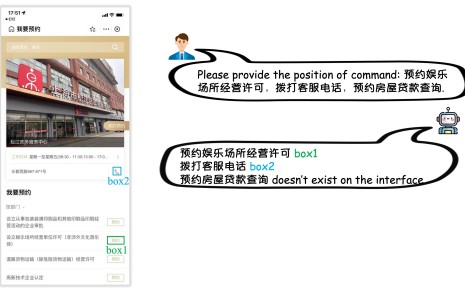
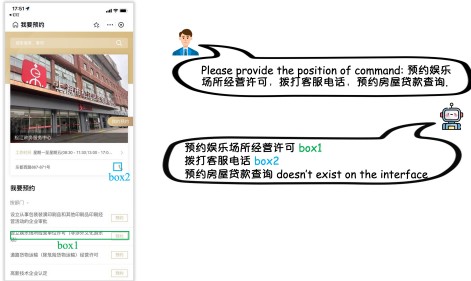

(c) For Chinese interface, Grounding GUI Anything achieves accurate multi-target grounding accuracy and rejection.

(d) For Chinese interface, while Qwen2.5-VL-7B achieved correct rejection, it has obvious grounding deviations.

Figure 8: The performance of our method and Qwen2.5-VL-7B on multi-target grounding with rejection mechanism.

### A.5 MORE DEMONSTRATIONS OF GROUNDING GUI ANYTHING

We present an analysis of existing basic model Qwen2.5-VL-7B (Bai et al., 2025) and Grounding GUI Anything with respect to grounding and parsing capability. We provide additional demonstrations of our method to further illustrate superior understanding and localization performance on multilingual interfaces with robustness and reliability, where the grounding task is shown in Figure 8, and the parsing task is shown in Figure 9.

### A.6 USE OF LLMS

In this work, we used OpenAI's GPT-5 solely for language polishing to improve the clarity, grammar, style and readability of the paper. The model was not involved in any core aspects of the research, including the motivation, novel ideas, and experimental design; all scientific contributions are entirely original.

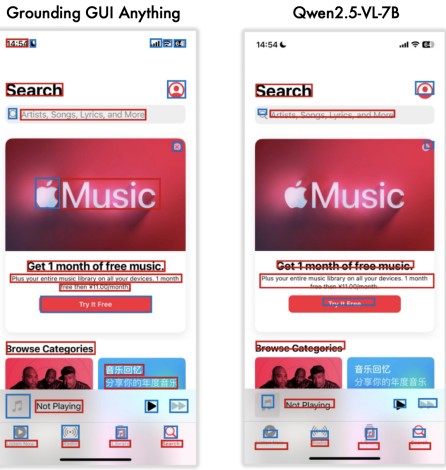
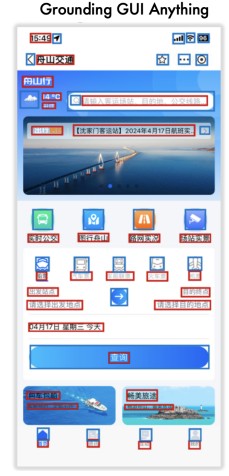
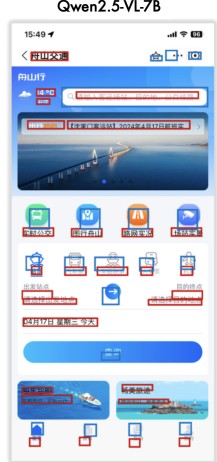

(a) Parsing results on English interfaces.  (b) Parsing results on Chinese interfaces.

Figure 9: For both Chinese and English interfaces, Grounding GUI Anything demonstrates the ability to accurately parse all elements, including icons and texts. In contrast, Qwen2.5-VL-7B exhibits certain limitations, including the omission of certain elements and locating deviations.

