# OpenReview forum: "Grounding GUI Anything: Efficient and Semantically-Aware Parsing via Continuous Coordinate Decoding"
_ICLR.cc/2026/Conference — ICLR 2026 Conference Withdrawn Submission_

### Official Review · Reviewer_Yrw3 · 2025-10-25

**Soundness:** 3
**Presentation:** 3
**Contribution:** 2
**Rating:** 2
**Confidence:** 5

**Summary:**

This work is mainly focused on the GUI grounding problem. The authors argued that the existing next-token prediction style bounding box prediction will constrain the grounding accuracy and proposed a dedicated regression-based decoder for element bounding box detection. A rejection mechanism was introduced to identify non-existent elements. They also proposed a semi-auto GUI element annotation pipeline to construct training datasets to enhance the structural perception ability of the model. Multiple experiments are presented to compare the proposed method to the existing ones.

**Strengths:**

GUI Grounding is one of the core challenges in the GUI Agent system and the research focus of this work is promising. The critique of using the next-token prediction paradigm for element bounding box prediction is interesting.

**Weaknesses:**

1. The literature review regarding GUI grounding tasks is incomplete, as it omits several key studies that employ non-next-token prediction paradigms. Specifically, the paper should cite and discuss:
  - Bbox regression-based GUI grounding:
    - GROUNDING MULTIMODAL LARGE LANGUAGE MODEL IN GUI WORLD (ICLR 2025).
  - Coordinate-free GUI grounding methods:
    - Attention-driven GUI Grounding: Leveraging Pretrained Multimodal Large Language Models without Fine-Tuning (AAAI 2025)
    - GUI-Actor: Coordinate-Free Visual Grounding for GUI Agents (NeurIPS 2025).
2. While GUI Grounding is a key step, the paper only provides performance metrics for this isolated component. The work currently lacks exploration into the system-level integration required to build a functional GUI Agent System. A demonstration of how this method seamlessly contributes to a unified and efficient full agent is necessary.
3. Figure 3 appears to show the output of the Vision Encoder being fed directly into the LLM, which contradicts the established architecture of InternVL 2.5, which typically includes a Pixel-shuffle layer and an MLP projector between the Vision Encoder and the LLM. The omission of this crucial intermediate module is confusing and raises several questions:
  - Was the MLP projector tuned in the proposed method?
  - Does the proposed coordinate generation module take the raw Vision Encoder output or the output after the MLP projector as its input?
4. What is the definition of $L_{sem}$ in Equation (4)?
5. Figure 5 utilizes Grounding DINO to detect icons within the GUI screenshots. It is suggested to include a quantitative analysis of the recall and precision of Grounding DINO specifically for this icon detection task on the proposed dataset. Furthermore, clarification is needed regarding the MLLM Semantics Module:
  - Is this module primarily dedicated to extracting icon semantics?
  - Which specific MLLM (model size and version) was utilized for this purpose?
  - What is the reported accuracy or quality assessment of the generated semantic response?
6. Could the authors clarify how the "Time" metric in Table 1 was calculated? Given that the model sizes being compared are comparable, the significant time difference seems disproportionate to the mere compression of a few bounding box output tokens. Is this time difference primarily due to a substantial variation in the overall output sequence length among the compared methods?
7. What is the exact formula used to compute the SemSim metric in Table 3? According to the table, the proposed GGA-8B model's performance on SemSim significantly surpasses that of Qwen 2.5-VL-72B. Is it because the GGA-8B model was exposed to related icons or semantic information from the proposed ScreenParse benchmark during its training phase, which might explain this large discrepancy?

**Questions:**

Please find the questions in the Weaknesses section.

---

### Official Review · Reviewer_W68Q · 2025-10-30

**Soundness:** 3
**Presentation:** 3
**Contribution:** 2
**Rating:** 2
**Confidence:** 4

**Summary:**

This paper proposes an end-to-end GUI grounding and parsing system named Grounding GUI Anything (GGA) that replaces traditional autoregressive coordinate token generation with a continuous regression-based decoder. It introduces a [VG] token to output bounding boxes directly and a [REJ] token to identify non-existent elements, aiming to improve both grounding precision and inference speed.

While the paper presents modest architectural improvements and strong empirical results, its contributions are not sufficiently novel or well-evaluated to meet the acceptance bar. Addressing evaluation breadth, better isolating the impact of components, and validating generalization to broader GUI scenarios would significantly strengthen the work.

**Strengths:**

- The paper presents a technically sound modification to existing GUI grounding approaches by replacing token-based coordinate generation with direct regression. The hallucination suppression through rejection tokens is also a useful addition.

- The authors also introduce a new benchamark, ScreenParse, for full GUI parsing.

- Results on ScreenSpot-v2, CAGUI-Grounding, and ScreenParse, often outperforming prior work, despite using fewer training samples.

**Weaknesses:**

- Limited Conceptual Novelty: The method is similar to adding a traditionally object detection head on the MLLM, it looks incremental and extra complexity to model architecture and training.  The rest of the architecture builds on standard MLLM pipelines with modest modifications (token routing, vision adapter).

- Evaluation Gaps: The empirical evaluation largely lacks results on end-to-end GUI reasoning performance and overall inference speed comparison on reasoning tasks. Will the extra grounding head influence the training of reasoning capability? We are unclear from current results.

Clarity Issues: Figures illustrating component contributions (e.g., ablation impact of [VG]/[REJ] tokens or element matcher) could be more explicit. Additionally, the inference time comparison is not clearly defined as end-to-end latency, leaving room for overinterpretation.

**Questions:**

N/A

---

### Official Review · Reviewer_Bqer · 2025-10-30

**Soundness:** 3
**Presentation:** 3
**Contribution:** 3
**Rating:** 8
**Confidence:** 4

**Summary:**

Grounding GUI Anything introduces a framework to improve grounding capabilities in GUI setups. Taking a pre-trained MLLM and its (internal) vision encoder, and LoRA finetuning it for GUI-specific tasks boosts performance in GUI-tasks (as shown in their ablations). Next to that, they further improve grounding by adding a VG token, which is replaced by a coordinate by the Coordinate Decoder -- not using autoregressive decoding for number prediction.

**Strengths:**

1. Shows great improvements without using too much compute,
2. Great figures that are clear + simple.
3. Good problem to tackle + simple solution.
4. Great evaluation setup, quite a few baselines + multiple benchmarks.

**Weaknesses:**

1. All figures look great, but figure 2 looks messy. Hard to see what to focus on and style is too different from the rest of the paper.
2. Token router seems to be explained as something complicated, but if I understand correctly it just extracts all tokens that are predicted and splits them into two groups: [VG] + [REJ] in group 1 (going to Coordinate Decoder) and all other tokens in group 2 (Going to Element Matcher + decode). From the text, it originally seemed like it was trained, or it did anything more. I would argue that in figure 3 you don't even need the extra "Token Router" block, since it's just a split (it's not similar to a token router in MoE).
3. In the tables, you mention "Time(s)", but it takes some time before you explain that's on "NVIDIA A100 (80G)" -- best to put that closer to the time mentioned.
4. "Average inference time per predicted element is measured using the "transformer" framework" -- can you add a reference to the transformer framework?
5. Table 5 is useful, but I miss two ablations:
    a. what is the impact of removing [REJ]?
    b. what is the impact of not LoRA finetuning the Vision Encoder? You mention it produces "enhanced visual features" by doing GUI-oriented localization, but you don't show anywhere that this is actually the case.

**Questions:**

1. I'm aware this is a hard ask because of compute limitations, but do you know how much it mattered that you picked "InternVL2.5-8B"? Given that you show it outperforms the other setups, could we go with a smaller model/setup and perform equally well to the previous sota?
2. What are the final weight losses you used?
3. How did you decide on the final architecture for the coordinate decoder? Again, this might be compute limited, but I assume no ablations on the architecture there have been done?
4. How much does the performance drop in non-GUI tasks? Ideally we could use this setup even in cases where we care about all original InternVL2.5-8B domains, but it's hard to predict how much the performance drops in the other cases (especially with the vision encoder finetuning only on GUI tasks).

---

### Official Review · Reviewer_itW5 · 2025-11-01

**Soundness:** 2
**Presentation:** 3
**Contribution:** 2
**Rating:** 4
**Confidence:** 4

**Summary:**

This paper proposes Grounding GUI Anything (GGA), a unified framework for grounding and understanding graphical user interfaces (GUIs). Unlike prior autoregressive methods that decode discrete coordinate tokens, GGA introduces a coordinate decoder with a specialized [VG] token for direct spatial regression and a [REJ] token for rejecting non-existent elements. The model integrates multimodal understanding through a route-then-predict architecture and is evaluated on ScreenSpot, ScreenSpot-v2, CAGUI-Grounding, and a newly constructed ScreenParse dataset. Experiments across multiple datasets demonstrate that GGA achieves superior grounding precision.

**Strengths:**

1. The paper identifies a concrete bottleneck in existing GUI grounding approaches, i.e., ineffective coordinate decoding, and proposes a unified framework that seeks to improve both accuracy and efficiency in practical scenarios.
2. The experiments cover multiple public benchmarks, providing solid evidence of the model’s performance and its effectiveness in suppressing hallucinations through the [REJ] token mechanism.

**Weaknesses:**

1. The technical novelty of the proposed seems incremental. The core idea is delegating coordinate prediction from next-token prediction to a decoder similar to those in object detection tasks. It is conceptually straightforward and not particularly insightful. The resulting performance improvement in grounding accuracy is therefore unsurprising. In fact, the more interesting and potentially novel contribution lies in the design of the [REJ] token, which helps suppress hallucinations, yet this part receives relatively little emphasis compared with the coordinate decoder itself.
2. Some concerns about efficiency gains. The paper repeatedly highlights efficiency as a major advantage, but the experimental results raise questions. In Tables 2 and 3, the proposed method is reported to be up to 5 times faster than the base model (InternVL2.5-8B), e.g., from 2.684 s to 0.632 s in Table 2, and from 0.558 s to 0.154 s in Table 3. This is counterintuitive, as an additional N-layer Transformer decoder (the coordinate decoder) is introduced. The paper does not clarify whether both methods are benchmarked on the same GPU setup or whether the reported time only measures the coordinate-output stage. If the latter, it would be useful to provide results for the overall framework latency, since the current claim of dramatic speedup is hard to interpret.

I have the above concerns; if the authors can address them clearly, I would consider raising my score.

**Questions:**

Please refer to the Weaknesses.

---

### Note · Authors · 2025-11-14

I have read and agree with the venue's withdrawal policy on behalf of myself and my co-authors.